# Fatigue Potentially Reduces the Effect of Transcranial Magnetic Stimulation on Depression Following COVID-19 and Its Vaccination

**DOI:** 10.3390/vaccines11071151

**Published:** 2023-06-25

**Authors:** Ayane Kamamuta, Yuki Takagi, Mizuki Takahashi, Kana Kurihara, Hibiki Shibata, Kanata Tanaka, Katsuhiko Hata

**Affiliations:** 1Tokyo TMS Clinic, Tokyo 150-0021, Japan; 2Department of Physics, Tokyo University of Science, Tokyo 162-8601, Japan; 3Department of Neuroscience, Research Center for Mathematical Medicine, Tokyo 183-0014, Japan; 4Department of Sports and Medical Science, Graduate School of Emergency Medical System, Kokushikan University, Tokyo 206-8515, Japan

**Keywords:** transcranial magnetic stimulation, long-COVID, fatigue

## Abstract

COVID-19’s long-term effects, known as Long-COVID, present psychiatric and physical challenges in recovered patients. Similarly, rare long-term post-vaccination side effects, resembling Long-COVID, are emerging (called Post-Vaccine). However, effective treatments for both conditions are scarce. Our clinical experience suggests that transcranial magnetic stimulation (TMS) often aids recovery in Long-COVID and Post-Vaccine patients. However, its effectiveness is reduced in patients with severe fatigue. Therefore, we retrospectively analysed Tokyo TMS Clinic’s outpatient records (60 in total; mean age, 38 years) to compare Long-COVID and post-vaccine patients’ characteristics and symptoms, assess the impact of TMS on their symptoms, and investigate the role of fatigue in depression recovery with TMS. The primary outcome was the regression coefficient of the initial fatigue score on depression score improvement using TMS. Secondary outcomes included psychiatric/physical scores before and after TMS and their improvement rates. We found no differences in the initial symptoms and background factors between Long-COVID and Post-Vaccine patients. After ten TMS sessions, all psychiatric and physical symptom scores improved significantly. TMS improves depression, insomnia, anxiety, and related neuropsychiatric symptoms, which were the primary complaints in this study. Thus, we conclude that TMS improves depression and anxiety. The effectiveness of TMS in treating depression in Long-COVID and Post-Vaccine patients decreased as fatigue severity increased. In conclusion, TMS relieved depressive symptoms following COVID-19 and vaccination; however, fatigue may hinder its effectiveness.

## 1. Introduction

The coronavirus disease (COVID-19) pandemic is raging and remains the most significant public health problem worldwide. The number of people with COVID-19 has increased along with the reporting of long-term symptoms (Long-COVID) in patients recovering from COVID-19. The symptoms of Long-COVID are diverse, with psychiatric and physical symptoms such as depression, poor concentration, anxiety, sleep disturbances, and fatigue, making daily life difficult for patients after recovery [1,2,3,4,5]. Nevertheless, effective treatments for Long-COVID have not yet been identified, and physicians treating Long-COVID patients are exploring various options.

Vaccination is essential to prevent the COVID-19 pandemic and has been carried out on a large scale worldwide. Recently, the long-term side effects of COVID-19 vaccination with a very low incidence have been reported [6,7]. These include many chronic conditions that occur later than neurological complications and thromboembolic/thrombocytopenic events that sometimes happen within one month of vaccination [8,9]. Therefore, diagnosis is difficult [7,10]. Only recently have the long-term side effects of the COVID-19 vaccine been reported, with an increasing number of reports suggesting that they may resemble the symptoms of Long-COVID [11,12]. However, there are very few reports on effective treatments for Long-COVID and the COVID-19 vaccine’s long-term adverse reactions.

Transcranial magnetic stimulation (TMS) is widely used to treat depression [13,14,15], bipolar disorder [16], obsessive compulsive disorder [17,18], anxiety [19,20], insomnia [21], and neurological rehabilitation [22,23]. To date, TMS has been performed in patients with psychiatric symptoms. Since January 2021, we saw a gradual increase in outpatients complaining of psychiatric and physical symptoms lasting more than a week, including depression, poor concentration, anxiety, sleep disturbances, and fatigue, after at least one week of SARS-CoV-2 infection (defined as Long-COVID patients). Similarly, since the summer of 2021, more outpatients have complained of these psychiatric/physical symptoms lasting more than approximately one week after at least one week of COVID-19 vaccination. Although diagnosing whether the COVID-19 vaccine caused these symptoms is difficult, it is true that the number of patients who complained that the COVID-19 vaccination caused their psychiatric/physical symptoms (defined as post-vaccine patients) has increased. In our TMS treatments of Long-COVID and Post-Vaccine patients since January 2021, we have observed recovery of such patients to the same degree as patients with typical depression, anxiety, and insomnia. Patients appear to recover better from depression if they are less fatigued. Indeed, an association between chronic fatigue and Long-COVID has been suggested regarding pathogenic mechanisms [24,25,26,27,28,29].

To scrutinise these clinical empiricisms, we conducted a detailed retrospective analysis of the medical records of Long-COVID and Post-Vaccine patients to verify the following:Are the symptoms of Long-COVID different from those of COVID-19-vaccine long-term adverse reactions?The effectiveness of TMS in the clinical presentation of these patients.Is fatigue involved in the TMS-induced recovery from depression in Long-COVID and Post-Vaccine patients?

Again, it remains unclear whether the long-term side effects of Long-COVID and COVID-19 vaccines are different and what effective treatments are available for them. We hope that the present study will further validate the efficacy of TMS in patients with symptoms of Long-COVID and COVID-19-vaccine long-term adverse reactions.

## 2. Methods

### 2.1. Study Design

This retrospective study used the medical records of patients who visited the Tokyo TMS Clinic. We compared the four psychiatric/physical symptom test scores before and after TMS treatment (before-and-after study). The involvement of a chronic fatigue indicator at the initial visit in the recovery rate on the depression scale was analysed using multivariate covariance analysis (MANCOVA).

### 2.2. Informed Consent for Patients

As part of the standard procedure, written informed consent was obtained from all patients before starting treatment in our clinic; the potential use of their anonymised medical records for research purposes, such as in this study, was thoroughly explained. This was in addition to explaining the expected benefits, costs, treatment duration, anticipated side effects, and measures to address the side effects of the treatment. Patient agreement to the use of their data was recorded as part of their written consent. We ensured that all patients were fully aware that their data could be used in this manner and had the right to withdraw their consent at any time without affecting their treatment.

### 2.3. Ethics Committee Approval

This study was conducted in accordance with the Declaration of Helsinki and approved by the BESLI CLINIC Ethics Committee (date of approval: 23 October 2022).

### 2.4. Transcranial Magnetic Stimulation Device and Coils

We performed transcranial magnetic stimulation using a Mag Pro R30 (Magventure, Denmark) connected to a circular coil (Magventure cool-125).

### 2.5. TMS Treatment

#### 2.5.1. Confirmation of Indications for Treatment

We asked the patients to complete a pre-interview form to ensure that the following absolute contraindications did not apply:
∙Age below 18 years∙History of head injury∙Hearing impairment∙Pregnancy or possible pregnancy∙Presence of metal (except titanium) near the stimulation site∙Cochlear implant or implantable neurostimulator∙Cardiac pacemaker∙Spinal cord or ventricle with a spinal fluid shunt∙Drug-infusion device∙Organic brain abnormalities found on MRI or CTFurther examination, as presented below, confirmed that the patient was ineligible for TMS treatment:∙History of psychiatric hospitalisation∙Strong feelings of hopelessness∙Strong verbal abuse, violence, and irritability∙Difficulties in communication∙Recommendations for inpatient treatment∙Schizophrenia∙Obsessive compulsive disorder∙Personality disorders∙Somatoform disorders∙Flashbacks∙Post-traumatic stress disorder∙Attachment disorders∙Physical and mental conditions for which treatment at a specialized medical institution is recommended∙Seizure disorders such as epilepsy and severe physical illnesses.

Patients who presented the above characteristics or whose doctors considered that they could not be safely treated were excluded.

#### 2.5.2. The Setting of the Stimulation Position

The physician selected the stimulation protocol during the first visit, based on the patient’s symptoms. The patients were asked to sit comfortably and fit matching-sized treatment caps. We aligned the centre of the hat with the midsagittal section and measured the distance from the nasal root (nasion) to the median anterior margin of the lid. This measurement ensured the reproducibility of the position of the head and cap after the second treatment by placing the cap such that the distance between the anterior outer edge of the lid and the nasal root matched that of the first time. Next, the international 10/20 method, which is standardised for the positioning of electroencephalography (EEG) electrodes, was used to measure the longitudinal (nasion to inion), transverse (tragus to tragus), and circumferential distances (circumference). Based on these values, the distance along the circumference from the midline (X) to the vertex (Y) was calculated using the BeanF3 method. Thus, we selected the dorsolateral prefrontal cortex (DLPFC) as the target of treatment [30,31].

#### 2.5.3. The Setting of Motor Threshold (MT Value)

The primary motor cortex of the abductor pollicis brevis muscle (APB) ipsilateral to the stimulation position was used as a reference. We defined the MT value as the minimum stimulus intensity at which the APB muscle contraction could be visually confirmed at least 5 out of 10 times by single-shot stimulation (5 Hz).

#### 2.5.4. Stimulus Intensity

After setting the stimulation position and motor threshold, the dorsolateral prefrontal cortex (DLPFC), the target of the treatment, was determined by gradually increasing the output from 50% of the motor threshold while confirming the patient’s pain (e.g., when the MT was 50, the stimulation intensity of 40 was 80% of the MT). If the patient experienced intense pain, the initial stimulation intensity was set to 60–80% of the MT value. The output increased as the number of treatments increased to 120% of the MT value indicated in the Guidelines for the Appropriate Use of Repetitive Transcranial Magnetic Stimulation (The Japanese Society of Psychiatry and Neurology).

#### 2.5.5. Treatment Initiation

At the initial visit, the patients were asked to sit in a comfortable position and align the centre of the cap used for measurement with the midsagittal section. We measured the distance from the nasal root (nasion) to the median anterior margin of the cap and matched it to the cap position at the initial visit. The patient was asked to wear earplugs to reduce stimulation sounds. The coil was set at the stimulus position and the treatment parameters were set.

#### 2.5.6. Assessment of Safety and Treatment Efficacy

The Quick Inventory of Depressive Symptomatology (QIDS) [32], Patient Health Questionnaire-9 (PHQ9) [33], Generalised anxiety disorder-7 (GAD7) [34], Performance Status (PS) [35,36], and other standardised scales were used to determine treatment efficacy, side effects, and future treatment strategies.

### 2.6. Data Collection

We selected the medical records of patients with post-COVID-19 sequelae and long-term COVID-19 vaccine side effects as the main complaints from approximately 2000 outpatients between 15 January 2021 and 29 September 2022 (Long-COVID group: 100 cases, Post-Vaccine group: 29 cases, total: 129 cases; Figure 1). We excluded patients who were judged to be off-label for treatment according to the Guidelines for the Appropriate Use of rTMS (Japanese Society of Psychiatry and Neurology) or who refused to accept treatment after an explanation of the TMS procedure ((Long-COVID group: 11 patients, Post-Vaccine group: 5 patients, total: 16 patients). We excluded patients who received TMS only for the first time because they only wanted to undergo the TMS procedure ((Long-COVID group: 11 patients, Post-Vaccine group: 5 patients, total: 27 patients). We excluded patients who discontinued treatment before the end of the 10 TMS sessions (between the 2nd and 10th sessions: Long-COVID group: 21 patients, Post-Vaccine group: 5 patients, total: 26 patients). Consequently, we included 46 patients with post-COVID-19 sequelae as the main complaint and 14 patients with long-term COVID-19 vaccine side effects as the main complaint, with medical records at the initial visit and after 10 TMS treatments. 

We defined the Long-COVID group as patients who, at the first visit, claimed to have psychiatric/physical symptoms applicable to the QIDS, PHQ9, GAD7, and PS lasting approximately one week or more after at least one week of COVID-19 and who thought that the disease had caused them. We defined the post-vaccine group as patients who claimed psychiatric/physical symptoms applicable to the QIDS, PHQ9, GAD7, and PS lasting approximately one week or more after at least one week of COVID-19 vaccination and who thought that the disease had caused them. Therefore, the present study did not conduct rigorous causal scrutiny of whether SARS-CoV-2 infection or the vaccination elicited symptoms.

Items extracted from the medical records included chief complaint, sex, magnetic stimulation intensity (% of motor threshold [MT]), rTMS protocol, medication use, duration from the first visit to 10 TMS treatments, and psychiatric/physical test scores (QIDS, PHQ9, GAD7, and PS) at the initial visit and after the tenth TMS treatment. Here, the chief complaint was the basis for determining whether the patient fell into the Long-COVID or Post-Vaccine group. The rTMS protocol included high-(left DLPFC) and low-frequency rTMS (right DLPFC).

### 2.7. Outcomes

The primary outcome was the partial regression coefficient of PS at the first visit in the MANCOVA, with the improvement rate of QIDS with TMS (ΔQIDS) as the dependent variable. The secondary outcomes were QIDS, PHQ9, GAD7, and PS scores at the initial visit and after 10 TMS procedures and their improvement rates.

### 2.8. Statistical Analysis

Among items extracted from the medical records, we defined patient background factors as follows: sex, age, TMS intensity, TMS protocol, the presence or absence of medication, and the number of days from the first visit to the 10 treatments. The Wilcoxon test was used to verify the difference in each background factor between the Long-COVID and Post-Vaccine groups. The QIDS scores ranged from 0 to 27, the PHQ9 from 0 to 27, the GAD7 from 0 to 21, and the PS from 0 to 9, with higher scores indicating more severe symptoms. We calculated the QIDS (%), PHQ9 (%), GAD7 (%), and PS (%) using Equations (1)–(4). We calculated the ΔQIDS, ΔPHQ9, ΔGAD7, and ΔPS through Equation (5). Note that X represents QIDS, PHQ9, GAD7, and PS. We excluded data with a psychiatric/physical symptom test score of zero at the first visit when calculating ΔX.
(1)$$\mathrm{QIDS}\left(\%\right)=\frac{\mathrm{QIDS}\;\mathrm{score}}{27}\times 100\;$$
(2)$$\mathrm{PHQ}9\left(\%\right)=\frac{\mathrm{PHQ}9\;\mathrm{score}}{27}\times 100\;$$
(3)$$\mathrm{GAD}7\left(\%\right)=\frac{\mathrm{GAD}7\;\mathrm{score}}{21}\times 100$$
(4)$$\mathrm{PS}\left(\%\right)=\frac{\mathrm{PS}\;\mathrm{score}}{9}\times 100\;$$
(5)$$\Delta X\left(\%\right)=\frac{X\left(\mathrm{After}\;10\;\mathrm{treatments}\right)}{X\left(\mathrm{Initial}\;\mathrm{vist}\right)}\times 100\;$$

We performed univariate analyses (Table 1 and Table 4) using the Wilcoxon test for significance. We performed univariate analyses (Table 2 and Table 3) and used the Wilcoxon signed-rank test to check for significance. When examining the multivariate correlations for each psychiatric/physical-symptom test score (Figure 2 and Figure 3), we determined the correlation coefficient, *p*-value of the correlation, and scatterplot matrix. We set the number of significant digits for each dataset to two decimal places and the *p*-value to one significant digit. The Restricted Maximum Likelihood (REML) estimation was used to handle missing data in multivariate correlation analysis.

We conducted single regression analyses and a MANCOVA to identify the initial visit score’s correlation with the primary outcome, ΔQIDS, secondary outcome, and the percentage improvement in PHQ9, GAD7, and PS. When carrying out the MANCOVA with ΔQIDS as the dependent variable, we used the four initial QIDS, PHQ9, GAD7, and PS as independent variables, along with information on whether the patient belonged to the Long-COVID or Post-Vaccine group, which was made into a dummy variable, with Long-COVID = 1, Post-Vaccine = −1.

To undertake a MANCOVA using propensity scores, we obtained a propensity score by logistic regression analysis with the independent variables of age, sex, medication status, TMS intensity, TMS protocol, and duration of treatment and the dependent variable of Long-COVID/Post-Vaccine. We then performed a MANCOVA on the independent variables of this propensity score plus the initial visit QIDS, PHQ9, GAD7, PS, and the dummy variable of Long-COVID/Post-Vaccine as the dependent variable of ΔQIDS (%). To avoid overfitting, we set the maximum number of independent variables for the MANCOVA to six [37].

We performed all statistics using JMP pro-version 15.0.0 (SAS Institute Inc., Cary, NC, USA), and statistical significance was set at *p* < 0.05 (two-tailed).

## 3. Results

In this study, we defined “Long-COVID patients” as outpatients who had complained of psychiatric/physical symptoms lasting more than about a week, such as depression, poor concentration, anxiety, sleep disturbances, and fatigue after at least one week of COVID-19. Similarly, outpatients who had complained of psychiatric/physical symptoms lasting more than about a week after at least one week of COVID-19 vaccination, such as depression, poor concentration, anxiety, sleep disorders, and fatigue, were defined as “Post-Vaccine patients”. Therefore, we did not conduct rigorous causal scrutiny of whether these symptoms were evoked by SARS-CoV-2 infection or vaccination. The psychiatric/physical symptom tests used in this study were the QIDS, PHQ9, GAD7, and PS. The QIDS and PHQ9 are indicators of depression [32,33] and GAD7 of anxiety disorder [34]. We used PS as an indicator of fatigue symptoms. PS is one of the reference indicators used in the criteria created by the Japanese Ministry of Health, Labour and Welfare (MHLW) based on the Myalgic Encephalomyelitis/Chronic Fatigue Syndrome (ME/CFS) diagnostic criteria, which were developed based on the requirements set by the Institute of Medicine (IOM), now the National Academy of Medicine (NAM) in the USA, in 2015.

First, we examined the background of Long-COVID and Post-Vaccine patients to determine which characteristics patients presented to the clinic and if there were any significant differences between the two groups. Table 1 shows no significant differences between the two groups regarding sex, age, TMS stimulation intensity, protocol, medication, or the number of days taken from the first visit to the 10th treatment.

**Table 1 vaccines-11-01151-t001:** Characteristics of the patients.

		Long-COVID (N = 46)	Post-Vaccine (N = 14)	*p* Value
Male (%)		31 (67.39)	6 (42.86)	-
Age (years old)		39.5 (29–47.25)	34 (25.25–42)	-
Stimulus intensity of TMS (% of MT)		91 (80–100)	84.5 (69.75–97.75)	-
Stimulation Protocol	High frequency rTMS (left DLPFC) (%)	41 (89.13)	13 (92.86)	-
	Low frequency rTMS (right DLPFC) (%)	4 (8.70)	0 (0.00)	-
	Other (%)	1 (2.17)	1 (7.14)	-
Medication (%)		28 (60.87)	12 (85.71)	-
Duration of treatment (days)		12 (7.75–19.5)	5.5 (3.75–22.75)	-

Data are presented as median (IQR) for continuous measures and n (%) for categorical measures.

We examined which items tended to be higher in each psychiatric/physical symptom test at the first visit in the Long-COVID and Post-Vaccine patients (Table 2). Because the highest score for each test differed, we normalised each test according to Equations (1)–(4); we then examined the significant differences between each psychiatric and physical-symptom test. We found that both Long-COVID and Post-Vaccine patients showed the same trend (Table 2), that is, both group patients showed significantly higher values for PHQ9 > GAD7 and PS > QIDS, in that order.

**Table 2 vaccines-11-01151-t002:** Differences in each psychiatric/physical symptom test score at the first visit.

	PHQ9 (%) − QIDS (%)	GAD7 (%) − QIDS (%)	GAD7 (%) − PHQ9 (%)	PS (%) − QIDS (%)	PS (%) − PHQ9 (%)	PS (%) − GAD7 (%)
N	46	46	46	39	39	39
Difference (%)	18.52 (7.41, 26.85)	5.29 (−1.98, 17.59)	−11.11 (−24.07, 2.91)	3.70 (−7.41, 25.93)	−14.81 (−22.22, 3.70)	3.17 (−9.52, 20.63)
*p* value	<0.0001 *	0.005 *	<0.0001 *	0.04 *	0.02 *	0.5
Post-Vaccine						
	PHQ9 (%) − QIDS (%)	GAD7 (%) − QIDS (%)	GAD7 (%) − PHQ9 (%)	PS (%) − QIDS (%)	PS (%) − PHQ9 (%)	PS (%) − GAD7 (%)
N	14	14	14	12	12	12
Difference (%)	22.22 (2.78, 34.25)	2.91 (−6.61, 23.94)	−8.99 (−23.81, 3.57)	25.93 (8.33, 50.93)	11.11 (−9.26, 28.70)	11.90 (−3.57, 49.60)
*p* value	0.002 *	0.2	<0.05 *	0.005 *	0.3	0.06

Data are presented as median (IQR). * Significant (*p* < 0.05).

Using the Wilcoxon signed-rank test, we examined how each psychiatric/physical symptom test score changed after ten sessions of TMS. The results showed that QIDS, PHQ9, GAD7, and PS scores improved significantly in both the Long-COVID and Post-Vaccine groups (Table 3). This was a retrospective before-and-after comparative study. However, numerous reports have shown that TMS effectively improves depression, insomnia, anxiety, and associated neuropsychiatric symptoms [13,14,15,19,20,21]. We also observed improvements in patients who underwent TMS. In the present study, Long-COVID and Post-Vaccine patients had varying degrees of these symptoms as primary complaints. Therefore, it is likely that the QIDS, PHQ9, and GAD7 scores improved with TMS. However, although positive results have been reported, the level of evidence concerning TMS in chronic fatigue is low [38]. Therefore, we cannot conclude from these results alone that the improvement in PS was due to TMS.

**Table 3 vaccines-11-01151-t003:** Changes in each psychiatric/physical symptom test score between the first.

Psychiatric Symptom		Number	Scores	*p* Value
QIDS (range 0–27)	Initial visit	46	6 (3–8)	<0.0001 *
	After 10 treatments	46	3 (1–5)	
PHQ9 (range 0–27)	Initial visit	46	10 (7–14)	<0.0001 *
	After 10 treatments	46	5 (3–9)	
GAD7 (range 0–21)	Initial visit	46	5 (3–10)	<0.0001 *
	After 10 treatments	46	3 (1–6)	
Physical Symptom				
PS (range 0–9)	Initial visit	39	2 (1–4)	0.01 *
	After 10 treatments	39	1 (1–3)	
Post-Vaccine				
Psychiatric Symptom		Number	Scores	*p* Value
QIDS (range 0–27)	Initial visit	14	7 (2.75–9)	0.04 *
	After 10 treatments	14	5.5 (1–7.5)	
PHQ9 (range 0–27)	Initial visit	14	14 (3.75–17.25)	0.0002 *
	After 10 treatments	14	9 (2.75–11.75)	
GAD7 (range 0–21)	Initial visit	14	5.5 (1–12.25)	0.001 *
	After 10 treatments	14	3.5 (0–7.25)	
Physical Symptom				
PS (range 0–9)	Initial visit	12	5.5 (2–6.75)	0.02 *
	After 10 treatments	12	4.5 (1.25–5.75)	

Scores are presented as median (IQR). * Significant (*p* < 0.05).

We conducted a univariate analysis of the initial psychiatric/physical-symptom test scores and the improvement rate for each score between Long-COVID and Post-Vaccine patients. To calculate the improvement rate, we divided the score after 10 TMS procedures by the score at the first visit, as shown in Equation (5) ΔX (%), where lower values indicated a better improvement rate. As shown in the “Initial Visit” row of each psychiatric/physical symptom test in Table 4, there were no significant differences in the scores between Long-COVID and Post-Vaccine patients at the initial visit. This result is similar to the conclusions presented in Table 2. Regarding the rate of improvement, only ΔQIDS differed between the two groups (row “ΔQIDS (%)” in Table 4). At first glance, Long-COVID patients seem to have a significantly better improvement rate in the QIDS than Post-Vaccine patients. However, the MANCOVA described below (Figure 3A–C) did not detect this difference. Thus, we found no significant differences in symptoms and improvement rates between Long-COVID and Post-Vaccine patients at the first visit (Table 2 and Table 4). These results are in agreement with our clinical hypothesis that the symptoms of Long-COVID and Post-Vaccine patients are very similar. Therefore, we assumed that the initial signs and their courses were the same in Long-COVID and Post-Vaccine patients.

**Table 4 vaccines-11-01151-t004:** Comparison of Long-COVID and Post-Vaccine patients on the initial value of each psychiatric/physical symptom test and the improvement rate with TMS treatment.

Psychiatric Symptom		Long-COVID (N = 46)	Post-Vaccine (N = 14)	*p* Value
QIDS (range 0–27)	Initial visit	6 (3–8)	7 (2.75–9)	-
	ΔQIDS (%)	50 (13.84–80.83)	83.77 (43.75–100)	<0.05 *
PHQ9 (range 0–27)	Initial visit	10 (7–14)	14 (3.75–17.25)	-
	ΔPHQ9 (%)	65.69 (41.25–100)	70.18 (48.61–88.90)	-
GAD7 (range 0–21)	Initial visit	5 (3–10)	5.5 (1–12.25)	-
	ΔGAD7 (%)	60 (33.33–100)	63.39 (25–85)	-
Physical Symptom				
PS (range 0–9)		Long-COVID (N = 39)	Post-Vaccine (N = 12)	*p* value
	Initial visit	2 (1–4)	5.5 (2–6.75)	-
		Long-COVID (N = 34)	Post-Vaccine (N = 11)	*p* value
	ΔPS (%)	93.75 (50–100)	83.33 (62.5–100)	-

Data are presented as median (IQR). * Significant (*p* < 0.05).

We investigated the degree of correlation between psychiatric/physical test scores at the initial visit and after TMS (Figure 2A,B). Unsurprisingly, the correlations between psychological scores (QIDS, PHQ9, and GAD7) were high at both the initial visit and after TMS. Interestingly, QIDS vs. PS correlated both at the first visit and after TMS, and PHQ9 vs. PS correlated after TMS. These results suggest that, in line with our hypothesis, fatigue may affect the recovery rate from Long-COVID and Post-Vaccine depression (hereafter referred to as “COVID-related depression”).

**Figure 2 vaccines-11-01151-f002:**
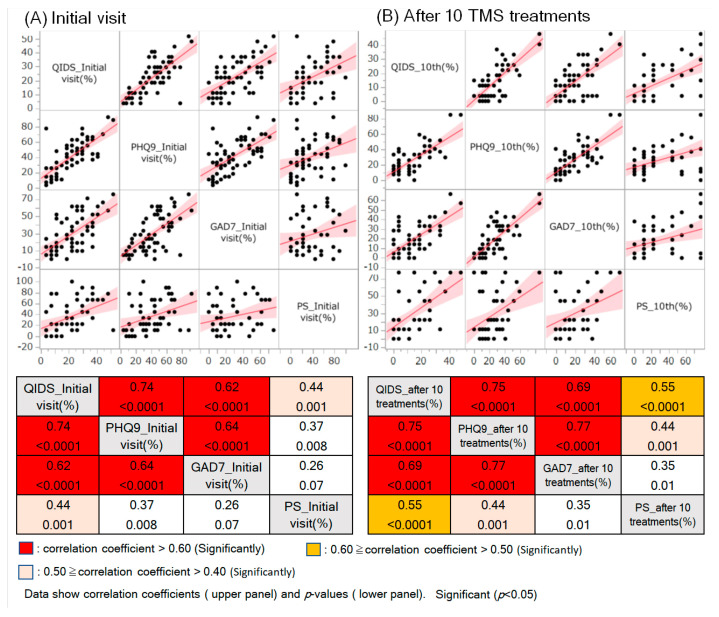
Correlations between each psychiatric/physical-symptom test score. Upper panels: Scatterplot matrix for each normalised psychiatric/physical test score. Values on the vertical and horizontal axes are expressed as percentages. Each figure shows the regression line and 95% confidence interval. Lower tables: The upper rows show the correlation coefficients between each normalised psychiatric/physical test score, and the lower rows show the *p*-value of the correlation coefficient; *p* < 0.05 is considered significant. Where the correlation coefficients were significant, we coloured them in three colours according to their magnitude.

We explored the impact of initial fatigue symptoms on the improvement rate of COVID-related depression using a single regression analysis and MANCOVA. In both studies, we found that ΔQIDS (%), the rate of improvement in the QIDS, was significantly positively correlated with PS (%) at the initial consultation (Figure 3A). No significant correlations were observed, except for this combination. We conducted a MANCOVA with ΔQIDS (%) as the dependent variable, each of the four initial visiting psychiatric/physical test scores as independent variables, and information about the group to which the patient belonged (Long-COVID/Post-Vaccine) as a dummy variable (Figure 3A: Model 1). PS (%) at the first visit showed a significant positive partial regression coefficient (Figure 3A). To further investigate the influence of PS (%) at the initial visit on ΔQIDS (%), we performed a MANCOVA with as many confounders as possible, recast as a single variable (propensity score). We obtained propensity scores using a logistic regression analysis with age, sex, medication status, the magnetic stimulation intensity, the magnetic stimulation protocol, and the duration of treatment (Table 1) as independent variables, and Long-COVID/Post-Vaccine as a dependent variable (Figure 3B). We conducted a MANCOVA with this propensity score as well as Long-COVID/Post-Vaccine, QIDS, PHQ9, GAD7, and PS at the first visit as independent variables, and ΔQIDS (%) as a dependent variable (Figure 3B: Model 2). Note that we considered it reasonable to include six independent variables in the present MANCOVA in reference to previous studies [37]. Similar to Figure 3A, PS (%) at the first visit showed a significantly positive partial regression coefficient, even after considering many confounding factors (Figure 3B). Finally, we adjusted for multicollinearity between independent variables and performed a MANCOVA. Models 1 and 2 show that the variance inflation factor (VIF) between the QIDS, PHQ9, and GAD7 at the first visit was approximately 2–3, as high correlation coefficients between these psychological test scores are also indicated in Figure 2. Thus, we conducted a MANCOVA of ΔQIDS (%) by selecting only QIDS scores from the psychological test scores as the independent variable and adding PS at initial diagnosis, Long-COVID/Post-Vaccine, and propensity scores from Model 2 as independent variables (Figure 3C: Model 3). The results also showed a significant positive partial regression coefficient for PS (%) at the initial consultation. As mentioned earlier in the explanation of Table 4, Long-COVID/Post-Vaccine did not significantly affect ΔQIDS (%). These results suggest that COVID-related depression is less likely to improve with TMS in patients with more pronounced fatigue symptoms.

**Figure 3 vaccines-11-01151-f003:**
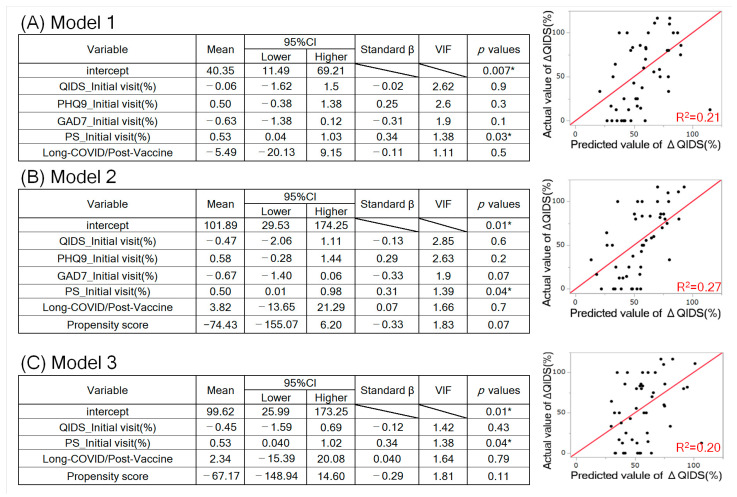
Multivariate analysis of covariance between ΔQIDS (%) and each psychiatric/physical-symptom test score at the first visit. (**A**–**C**) MANCOVA results for each of the three patterns. The partial regression coefficients of PS_Initial visit (%) are significantly positive in all models. In Model 3 (**C**), the VIF is below 2. In each right-hand diagram, the horizontal axis shows the predicted value of ΔQIDS (%) obtained by adding the intercept and all values obtained by multiplying each independent variable by its partial regression coefficient. The vertical axis represents the measured value of the ΔQIDS (%). * Significant (*p* < 0.05).

## 4. Discussion

The present study is the first to elucidate the validity of TMS and the factors underlying its use for treating psychiatric symptoms after COVID-19 and vaccination. We found no difference in the characteristics or initial symptoms between patients who presented with psychiatric/physical symptoms after COVID-19 (Long-COVID patients) and those who presented with psychiatric/physical symptoms after COVID-19 vaccination (Post-Vaccine patients). Comparing the QIDS, PHQ9, GAD7, and PS scores before and after TMS treatment in both groups, all items significantly improved. There were no significant differences between the groups in the rates of improvement of QIDS, PHQ9, GAD7, and PS with TMS. We, therefore, assumed that the initial symptoms and course of Long-COVID and Post-Vaccine patients had been the same and defined their depression as “COVID-related depression”. In addition, although this was a retrospective before-and-after study, numerous reports have suggested that TMS improves QIDS, PHQ9, and GAD7. Stimulating the DLPFC with TMS has been shown to induce neuroplasticity in this underactive region, improving depression and anxiety [39,40,41,42]. Many studies in patients with depression have shown significant rTMS-induced changes in functional connectivity between areas important for emotion regulation, including the DLPFC and the subgenual anterior cingulate cortex, and among the default mode, salience, and central executive networks [43]. Thus, we attributed the improvements in the QIDS, PHQ9, and GAD7 scores to TMS. Meanwhile, our results alone made it difficult to conclude that TMS improved PS. We then explored the influence of fatigue on the improvement rate of COVID-related depression by conducting a single regression analysis and MANCOVA. We found that the higher the PS at the initial visit (i.e., the stronger the chronic fatigue symptoms), the worse the COVID-related depression recovery rate.

Long-COVID and Post-Vaccine patients presented with significantly higher psychiatric/physical symptom values at their first visit in the order of PHQ9 > GAD7 and PS > QIDS, and similar trends were observed (Table 2). Depression, poor concentration, sleep disturbances, anxiety, and fatigue were common to both groups of patients who visited the Tokyo TMS Clinic. The recovery rate for each symptom did not differ between the two groups (Table 4). COVID-19 has been reported to cause severe inflammatory symptoms, and even minor infections are associated with cytokine elevation and brain microglial activation that persists for a long time [27]. In contrast, the COVID-19 vaccine, including mRNA vaccines distributed in Japan (mRNA-1273 SARS-CoV-2 vaccine and BNT162b2 mRNA COVID-19 vaccine [44,45]), is known to cause Long-COVID-like symptoms, albeit at a low rate [11]. Reports indicated that COVID-19 vaccines caused severe inflammatory symptoms by disrupting innate immunity, suppressing IFN-α signalling, failing to prevent and detect intracellular malignant transformation, and generating large numbers of exosomes carrying spiked glycoproteins [12]. Consequently, it can be inferred that widespread inflammation is involved in the onset mechanisms of both Long-COVID and the post-vaccination long-term side effects of COVID-19. Furthermore, the involvement of anti-idiotype antibodies (Ab2) is a common mechanism in both conditions [46]. Ab2 reacts with a specific antibody (Ab1) against an antigen, and Ab2’s antigen-binding region mimics the original antigen. This allows Ab2 to bind to the same receptor targeted by the original antigen, potentially affecting the cell and causing pathological changes, even after the original antigen disappears. This mechanism could explain why the symptoms of Long-COVID and the post-vaccination long-term side effects of COVID-19 are similar. Suppose that Ab1 generated by the COVID-19 vaccination is an antibody against the spike protein of SARS-CoV-2. In this case, it is possible that Ab2, which can bind to the widely expressed ACE2 receptors on the spike protein, is produced. Meanwhile, Ab1 is also produced after SARS-CoV-2 infection; therefore, Ab2, which is capable of binding to ACE2, can be produced similarly. Therefore, in post-SARS-CoV-2 infection and post-COVID-19 vaccination, specific Ab2 can regulate the function of ACE2, leading to widespread inflammation. It can lead to various neurological symptoms, as inflammatory cytokines cause psychiatric and physical symptoms, such as depression, insomnia, anxiety, and chronic fatigue [47,48,49,50,51,52]. Thus, it can be inferred that the onset mechanisms of Long-COVID and Post-Vaccine patients in this study have many commonalities.

Even though the QIDS and PHQ9 are highly sensitive and specific diagnostic criteria for depression, both Long-COVID and Post-Vaccine patients tended to score significantly higher on the PHQ9 than on the QIDS at the first visit [32,33]. It is necessary to conduct further research on whether this difference depended on how the QIDS and PHQ9 questions were asked or on the symptoms of the Long-COVID/Post-Vaccine patients.

After 10 TMS treatments for patients presenting with psychiatric/physical symptoms, including depression, triggered by COVID-19 and COVID-19 vaccination, there was a significant improvement in all test scores compared to the pre-treatment scores (Table 3). From this before-and-after study alone, it is difficult to conclude whether TMS ameliorates the psychiatric and physical symptoms. However, several reports have demonstrated the therapeutic effects of TMS on depression, insomnia, anxiety, and the associated neuropsychiatric symptoms [13,14,15,19,20,21]. The patients in this study who claimed that COVID-19 or COVID-19 vaccination were the cause also had chief complaints of psychiatric symptoms, including depression, poor concentration, insomnia, and anxiety. The depression scales QIDS and PHQ9 used in this study included questions about depression, poor concentration, and insomnia. GAD7 is a diagnostic indicator of the degree of generalised anxiety disorder. Therefore, it is natural to consider that TMS improves QIDS, PHQ9, and GAD7 [32,33,34]. However, there is limited medical evidence regarding the effectiveness of TMS in treating chronic fatigue [38]. In this study, although TMS significantly improved PS compared to pre-treatment, we cannot rule out the possibility that PS could have improved spontaneously, even without TMS. Whether TMS improves fatigue symptoms remains to be further explored in a double-blind, randomised controlled trial.

We found using MANCOVA that the higher the PS (used as a reference measure for ME/CFS diagnosis) at initial diagnosis, the worse the improvement rate of the QIDS. While we may not be able to conclude that ME/CFS is based solely on high PS, our findings align with our clinical observations, indicating that chronic fatigue could potentially impact recovery from COVID-related depression. The association between chronic fatigue and COVID-related depression has also been indicated in terms of both the pathogenic mechanisms. Patients with ME/CFS had higher levels of oxidative stress and more widespread microglial activation in the brain than healthy subjects [24,25]. Interestingly, the level of microglial activation in patients with ME/CFS positively correlated with cognitive impairment and the severity of depression and pain. In contrast, autopsy reports of patients who died from COVID-19 reported the activation of microglia in the brain when SARS-CoV-2 infection in the brain was not confirmed [26]. Experimental studies in mice with SARS-CoV-2 infection confined to the lungs have also revealed microglial activation in the white matter [27]. A recent study found that Long-COVID patients are subjected to higher oxidative stress than healthy subjects [28]. These findings suggest that activation of microglia in the brain and an increase in oxidative stress evoke ME/CFS and Long-COVID. Brain inflammation and high oxidative stress in patients with high PS levels may be associated with resistance to TMS treatment. Controlling inflammation and oxidative stress with TMS in such patients is necessary for recovery from COVID-related depression with TMS [29].

The clinical and theoretical implications derived from this research are as follows:Effectiveness of TMS: This study provides empirical evidence supporting the effectiveness of TMS in treating psychiatric symptoms following COVID-19 and its vaccination. It expands our understanding of the potential applications of TMS in the context of pandemics.Similarity of Long-COVID and Post-Vaccine Symptoms: The study suggests that the psychiatric symptoms and their progression in Long-COVID and Post-Vaccine patients are similar, leading to the concept of “COVID-related depression”. It could prompt further research into the common underlying mechanisms of these conditions.Role of Chronic Fatigue: The finding that stronger chronic fatigue symptoms were associated with a worse recovery rate from COVID-related depression suggests that chronic fatigue may be an independent exacerbating factor for COVID-related depression. This could stimulate further research on the interplay between fatigue, depression, and other psychiatric symptoms in the context of COVID-19 and its vaccination.Limitations of TMS: This study suggests that while TMS may improve depressive and anxiety symptoms, it may not necessarily improve fatigue symptoms. This could prompt further investigations into the limitations of TMS and the need for additional or alternative treatments for fatigue in these patients.

These implications could guide future research in this area by informing the development of more effective treatment strategies for patients with psychiatric symptoms after COVID-19 and vaccination. They also highlight the need for a more nuanced understanding of the role of chronic fatigue in the progression and treatment of COVID-related depression, potentially leading to the exploration of additional or alternative treatments for fatigue. Ultimately, these insights could improve patient outcomes and quality of life after the COVID-19 pandemic.

The limitations of this study were as follows. A before-and-after comparative study showed improvements in psychiatric and physical symptoms with TMS. We could not test which TMS stimulation protocols resulted in recovery from COVID-related depression, as the physician selected the stimulation protocols at the first visit according to the patient’s symptoms. Most stimulation protocols involved high-frequency stimulation of the left DLPFC. We could not rule out the possibility that uncorrected and unknown confounding factors may have influenced the MANCOVA results.

## 5. Conclusions

This study provides critical insights into the psychiatric symptoms following COVID-19 and its vaccination, collectively termed “COVID-related depression”. Our findings suggest that Long-COVID and Post-Vaccine patients exhibit similar psychiatric symptoms and progression and that both groups significantly benefit from TMS treatment. However, chronic fatigue appears to negatively influence the recovery rate from COVID-related depression, indicating the need for further investigation into the role of fatigue under these conditions. Although TMS shows promise in treating depressive and anxiety symptoms, its effectiveness in alleviating fatigue remains unclear. These findings underscore the need for continued research on the underlying mechanisms of Long-COVID and Post-Vaccine symptoms, the role of fatigue in these conditions, and the exploration of additional or alternative treatments for fatigue. Ultimately, these insights could lead to more effective treatment strategies, improving patient outcomes and quality of life after the COVID-19 pandemic.

## Figures and Tables

**Figure 1 vaccines-11-01151-f001:**
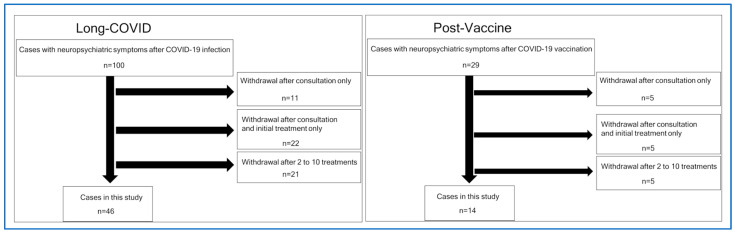
List of target data and exclusions.

## Data Availability

All data generated in this study are available if the BESLI CLINIC Ethics Committee considers it a reasonable request.

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
