# Peer review of "Fatigue Potentially Reduces the Effect of Transcranial Magnetic Stimulation on Depression Following COVID-19 and Its Vaccination"

_vaccines, 2023, doi:10.3390/vaccines11071151_

Round 1

Reviewer 1 Report

Title: title is unclear, needs to be amended

Please provide underlying mechanism proposed hypothesis

The study is retrospective using medical records, how Informed consent for patients were obtained for current analysis

Please present the methodology of the manuscript according to secondary data analysis

Please provide clinical and theoretical implications of the manuscript

 Moderate editing of English language required

Author Response

Thank you very much for your very useful comments. We believe that your comments have improved our manuscript very well. If further corrections are needed, please do not hesitate to point them out to us. Thank you very much.

Our common comments to reviewers 1 and 2 are as follows.
1. as p-values were written in different formats in the manuscript and figures (e.g. p<0.05, P<0.05), we chose one style (e.g. p<0.05) and applied it consistently throughout the text.
2. edited 'long-covid' to 'long-COVID' in the manuscript, all tables and figures
3. please note that the English wording has changed compared to the first edition, as we asked an English proofreader to revise the English language of the text throughout the manuscript.

For other comments on each Reviewer, please see the attached file.

Reviewer 2 Report

-      Dear authors, I'm glad to have the opportunity of reviewing your manuscript, which has relevant topic for public health. Manuscript has its’ strengths, however, there are my comments that I believe would help to increase the quality of the manuscript.

   Abstract: The abstract does not follow the editorial standard indicated for IJERPH: “The abstract should follow the style of structured abstracts, but without headings”; therefore, remove the words: Background, Methods, Results and Conclusions.

-        English needs extensive editing to facilitate ease of reading.

-   The discussion is too short and confusing to highlight the research gap. The paper needs to be carefully reorganized with a clearly stated argument spelled out in the closing pages. Where are the Conclusions?

    Use past tense when discussing the procedure and results as well as other researchers' procedures and results.

The abstract should provide information about study composition (number of patients, age).

  English needs extensive editing to facilitate ease of reading.

Author Response

(The authors gave the same response as above.)

Round 2

Reviewer 1 Report

The manuscript has been revised and improved.